# Physiological Health and Physical Performance in Multiple Chemical Sensitivity—Described in the General Population

**DOI:** 10.3390/ijerph19159039

**Published:** 2022-07-25

**Authors:** Anne A. Bjerregaard, Marie W. Petersen, Sine Skovbjerg, Lise K. Gormsen, José G. Cedeño-Laurent, Torben Jørgensen, Allan Linneberg, Thomas Meinertz Dantoft

**Affiliations:** 1Center for Clinical Research and Prevention, Bispebjerg and Frederiksberg Hospital, Nordre Fasanvej 57, Hovedvejen, Entrance 5, 2000 Frederiksberg, Denmark; torben.joergensen@regionh.dk (T.J.); allan.linneberg@regionh.dk (A.L.); thomas.meinertz.dantoft@regionh.dk (T.M.D.); 2Research Clinic for Functional Disorders and Psychosomatics, Aarhus University Hospital, Universitetsbyen 22–23, 8000 Aarhus C, Denmark; mawept@rm.dk (M.W.P.); lisgor@rm.dk (L.K.G.); 3The Danish Center for Mindfulness, Department of Clinical Medicine, Aarhus University, Hack Kampmanns Plads, 8000 Aarhus C, Denmark; sine.skovbjerg@clin.au.dk; 4Department of Exposure Epidemiology and Risk Program, Harvard T.H. Chan School of Public Health, 1350 Massachusetts Ave, Cambridge, MA 02138, USA; jcedenol@hsph.harvard.edu; 5Department of Public Health, Faculty of Health and Medical Sciences, University of Copenhagen, Blegdamsvej 3B, 2200 Copenhagen, Denmark; 6Department of Clinical Medicine, Faculty of Health and Medical Sciences, University of Copenhagen, Blegdamsvej 3B, 2200 Copenhagen, Denmark

**Keywords:** multiple chemical sensitivity, MSC, DanFunD, functional somatic disorders

## Abstract

Multiple chemical sensitivity (MCS) is a multifactorial somatic disorder characterized by physical reactions triggered by even extremely low levels of different airborne chemicals. In most individuals with MCS, these reactions have substantial negative impact on social, occupational, and everyday life often including limited or no engagement in physical activities. The aim of this study was to explore associations between MCS and objective measurements of anthropometry, cardiorespiratory health, and physical performance. From the Danish population-based cohort DanFunD counting 9656 participants aged 18–76 years, 1.95% (*n* = 188) were categorized as MCS individuals (MCS All). Of those 188, 109 participants were subcategorized as having MCS without functional somatic disorders (FSD) (MCS with no comorbid FSD). The remaining study population without any FSD were regarded controls. We used adjusted multiple linear regression analyses to evaluate associations between MCS and anthropometry, cardiorespiratory fitness, and physical performance. Compared with the general population, MCS All had less optimal body composition, increased risk of obesity, impaired cardiorespiratory fitness, and physical performance which was not seen in MCS with no comorbid FSD. MCS individuals may be inhibited to maintain an active lifestyle which can increase risk of obesity and consequently have negatively impact on general health, which may not be the case among MCS with no comorbid FSD.

## 1. Introduction

Individuals with multiple chemical sensitivity (MCS), a functional somatic disorder (FSD) [1], is characterized by physical symptoms triggered by even extremely low levels airborne chemicals such as odors, perfumes, car exhaust etc. This often results in symptoms such as headache, fatigue, nausea, muscle, and joint pain and involves symptoms from several organ systems e.g., the respiratory or the cardio-vascular system and most frequent the central nervous system [2,3]. MCS individuals often suffer from comorbid FSD including chronic fatigue syndrome, fibromyalgia, and/or irritable bowel syndrome [4]. There is currently no treatment of MCS but to avoid being exposed or removal from symptom triggering chemicals.

The absence of a clear case definition criteria influences the prevalence estimate of MCS which ranges from 0.5% based on physician diagnosis [3] to 12.6% based on telephone-based interviews [5]. The Danish Study of Functional Disorders “DanFunD”, population-based cohort study (*n* = 9656), presents a prevalence of 1.95% [6] whereas a study including a sample of 4435 adults representative on age, sex, and region from the United States, Australia, United Kingdom, and Sweden reported a prevalence of 7.4% on average [7]. Because MCS is poorly captured in registries, questionnaires are the most used tool to assess MCS, however, these may vary in number of questions as well as content [5,8,9,10].

The consequences of MCS can be substantial, including negative impact on social and occupational life as well as everyday life such as limited or no engagement in physicals activities [6,11,12]. The association between low level of physical activity and long-term health is well established [13]. Even among individuals with no other known risk factors, low level of physical activity has been associated with increased risk of heart disease [14], type 2 diabetes [15], and obesity [16]. However, few studies have investigated cardiorespiratory health and physical performance based on objective measurements in MCS individuals. Population-based studies found MCS to be associated with physical inactivity, poor quality of sleep [6], and increased respiration and heart rate [17], whereas a smaller clinical study found low muscle strength and muscle mass in MCS patients [18]. MCS symptoms overlap with those of other hypersensitivity syndromes such as asthma and previous studies in general populations have also reported associations between MCS and asthma [5,9,19,20,21], but only few epidemiological studies have supported such observations by including objective, clinical data.

Objective measurements of cardiorespiratory health and physical performance in MCS individuals seems sparse and has to our knowledge not previously been investigated in comparison with the general population. The aim of the paper was to further elucidate whether MCS is associated with poor physiological health in terms of decreased physical performance, low cardiorespiratory fitness, and less optimal body composition. Such insights can contribute to focused health promotion among MCS individuals and patients.

## 2. Materials and Methods

### 2.1. Study Population

DanFunD have been described in details elsewhere [22]. In short, during 2011–2015 28,773 men and women aged 18–76 years, living in 10 municipalities in the Western part of greater Copenhagen were randomly selected via national registries and invited to participate in the DanFunD study. All participants completed a general health examination and filled in an extensive an extensive questionnaire. All participants were examined between February 2012 and June 2015 on weekdays between 7 am and 3 pm and total length of each examination was about 1.5 h.

Case criteria for MCS were constructed as a reduced adaptation of the 1999 US Consensus Criteria for MCS and the revisions suggested by Lacour and colleagues [23,24]. Thus, three criteria should be fulfilled: (1) having experienced symptoms of at least two of 11 common odors exposures, (2) presence of at least one symptom from the central nervous system and at least one symptom from another organ system, and (3) symptoms cause significant changes in lifestyle [25]. From the full study population, based on standardized validated questionnaires, participants fulfilling criteria for MCS [23,24] were identified. Similarly, four types of comorbid FSDs were delimited [26] i.e., chronic widespread pain [27], chronic fatigue [28,29], irritable bowel [30], and whiplash-associated disorders [31]. For analyses, we divided the study population into two primary study groups and one MCS subgroup, i.e.,
MCS All: All participants fulfilling MCS criteria allowing presence of one or more comorbid FSDs.MCS with no comorbid FSD: All cases fulfilling MCS criteria not allowing comorbid FSDControls: All cohort participants not fulfilling criteria’s for any FSD.

No exclusion criteria were applied.

### 2.2. Outcome Assessment

All outcomes were assessed at a health examination at Center for Clinical Research and Prevention, Copenhagen, carried out by five experienced research nurses. For all outcomes, standardized and detailed research protocols were prepared and reproducibility between staff and across the study period was monitored by periodical internal validations by coworkers.

Anthropometrics included one measurement of waist circumference (cm) by placing the measuring tape midway between the lower rib curvature and the upper hip edge. We applied cutoff from the American Heart Association to estimate the proportion of individuals above/below optimal waist circumference which is 102 cm for men and 88 cm for women [32]. Weight and body fat percentage was measured once with light clothes (fasting) using a Tanita TBF-300 body composition analyzer. BMI (kg/m^2^) was calculated based on height (without shoes) and weight.

Cardiorespiratory fitness was assessed by spirometry (spiropharma.dk) and included forced vital capacity (FVC, L) which is a measure of the amount of air one person can forcefully exhale from the lungs, and forced expiratory volume in the first second (FEV_1_, L) which is a measure of the amount of air one person can forcefully exhale from the lungs in the first second. Both measures are highly correlated with age, sex, height, and ethnicity [33] which were taken into consideration via adjustment in all analyses except for ethnicity. In addition, the FEV1/FCV ratio was calculated which is a measure used for evaluating obstructive lung conditions often based on a cutoff of 70% for mild obstruction [34]. Finally, systolic, and diastolic blood pressure (BP) (mmHg) was assessed in sitting position with a mercury manometer after 5 min of rest with two repetitions one minute apart. The mean of two measurements were used in analyses.

Physical performance was evaluated based on a step test, a hand grip test, and level of self-perceived fitness. The step test included a gradually increasing performance-limited max test using a step bench (20–35 cm) to estimate metabolic equivalents (METs) [35]. The hand grip test was performed using a dynamometer (JAMAR^®^, pounds) with three repetitions in both right and left hand. The dominant hand was registered and the mean of the two first repetitions were used in analyses. Self-perceived fitness [36] was assessed with a single questionnaire item:” How do you consider your own physical fitness?” on a 5-point Likert scale ranging from “really good” to “bad”. The scale was dichotomized combining “really good”, “good”, “acceptable” and “less acceptable”, and “bad”. For both asthma, obstructive pulmonary disease (OPD), and hypertension, self-reported diagnosis by a medical doctor was retrieved from one questionnaire item asking “Have you ever been told by a physician that you suffer from..?”.

### 2.3. Covariate Assessment

Information on other lifestyle factors and mental vulnerability was collected via validated questionnaires. This included smoking habits (dichotomized for analyses: daily/regularly and former/never), number of drinks per week (dichotomized for analyses: below ≤21 for men and ≤14 for women) [37], and sleep disturbances (dichotomized for analyses: daily/almost daily and weekly/monthly/rarely). Both sleep-disturbances and self-perceived health (30) was measured with a single question on a 5-point Likert scale. Self-perceived health ranged from good to poor and was dichotomized into ‘poor’ (fair/poor) and ‘good’ (excellent/very good/good) for analyses. Self-perceived stress was assessed using Cohen’s Perceived Stress scale [38] ranging from zero to 40 and mental health including anxiety and depression was assessed using the anxiety and depression subscales from the Symptom Checklist 90 [39].

### 2.4. Statistical Analyses

Statistical analyses were performed using SAS Enterprise guide 7.15. Results were assessed at a 5% significance level.

Standard descriptive statistics (mean (± standard deviation (SD))/median (percentiles)/*n* (%)) was applied to describe distributions of outcomes and covariates within MCS, MCS with no comorbid FSD, and controls, respectively. To test mean/median difference between MCS or MCS with no comorbid FSD and controls, *t*-test (one-way ANOVA) or Kruskal-Wallis test was applied for continuous variables with normal or skewed distribution, respectively. The chi squared (*X*^2^) test was applied for categorical variables.

Multiple linear regression analyses using the SAS Genmod procedure was conducted to evaluate associations between anthropometry, cardiorespiratory fitness, and physical performance as dependent variables and MCS or MCS with no FSD comorbidities compared with controls as independent variable. Three models were applied: model 1 adjusted for age and sex; model 2 additional adjusted for lifestyle factors including alcohol, smoking, and sleep disturbances; model 3: additionally, adjusted for mental health factors including Cohens perceived stress scale, depression, and anxiety. Lung function measures (FVC and FEV1) was also adjusted for height in all models. To test effect modification of chronic stress, interaction between MCS or MCS with no comorbid FSD and Cohens perceived stress scale was examined in model 3 for all outcomes. In ad hoc analyses, we explored whether antihypertension treatment could explain observed significant associations between MCS and BP, and whether OPD or asthma could explain significant associations between MCS and lung function measures. This was done by additionally adjusting model 3 for self-reported treatment for hypertension (yes/no), self-reported OPD diagnosed by a medical doctor (yes/no), or self-reported asthma diagnosed by a medical doctor (yes/no), respectively. To test whether the model assumptions were met, residuals for normal distribution, homogeneity of variance, and interaction was examined between independent variables and age and sex, respectively. The best model fit was tested by Hosmer and Lemeshow and achieved by including age to the power of two. Thus, for some variables age seemed to have a different importance (for example on blood pressure and hand grip in MCS). Associations are presented as beta-coefficients and 95%CI.

## 3. Results

A total of 188 persons fulfilled the criteria for MCS, and of those, 109 only fulfilled criteria for MCS only with no additional FSD. Those participants were assigned to the MCS with no comorbid FSD subgroup. Of the remaining cohort 7791 did not fulfil criteria for any of the five FSD´s and were regarded controls.

Population characteristics are presented in Table 1. Mean age were compareble among controls, MCS All, and the MCS with no comorbid FSD subgroup and compared with controls, the proportion of women was significantly higher in both MCS All and in the MCS with no comorbid FSD subgroup. MCS All individuals had significantly less optimal body composition and cardiorespiratory fitness compared with controls except for diastolic BP. Whereas for the MCS subgroup without FSD, participants only differed from controls on a few parameters i.e., higher fat percentage as well as lower hand grip strengths, lower lung capacity, and lower step METs count. Both MCS individuals with and without FSD comorbidities self-reported lower mental health compared with controls whereas for other life-style factors, MCS All more often reported sleep disturbances and MCS with no comorbid FSD more often reported problems waking up early compared with the controls. Less women among MCS with no comorbid FSD than controls had an alcohol intake below recommended levels.

From the regression analyses, we found a significantly higher waist circumference, fat percentage, and BMI in MCS All compared with controls which was not explained by the included confounding lifestyle and mental health factors (Table 2). There were increased odds of being overweight/obese among MCS All compared with controls (Figure 1). When restricting analyses to MCS individual without FSD, no significant associations were seen between anthropometry measures compared with controls (Table 2 and Figure 1).

For cardiorespiratory fitness, the FVC and FEV1 was significantly negatively associated with MCS All, but not MCS with no comorbid FSD (Table 3). Further adjustment for OPD (yes/no) or asthma (yes/no) in MCS did not change the observed association of decreased lung function compared with controls (data not shown). No significant associations were observed for the FEV1/FVC ratio (Table 3). Due to the self-reported nature of OPD and asthma diagnoses these subgroups were described in ad hoc analyses. Thus, the 9% MCS All individuals who reported to have OPD (mean [SD] age: 61.9 [7.8] years), 76% (*n* = 13) reported to be occasionally or daily smokers. In the control group, 2% reported to have OPD diagnosed by a medical doctor (mean [SD] age 62.5 [8.8] years) of whom 72% (*n* = 99) were former, occasionally, or daily smokers. In MCS All individuals, 26% (*n* = 49) self-reported asthma diagnosed by a medical doctor which was significantly more compared with the controls 9% (*n* = 710) (*p* < 0.001).

Systolic BP was significantly lower both among MCS individuals with and without FSD comorbidities compared with the controls which could not be explained by the included confounding factors (Table 3). Number of individuals who reported treatment for hypertension were 20%, 24%, and17% among controls, MCS All, and MCS with no comorbid FSD, respectively. Both in MCS individuals with and without FSD comorbidities, further adjustment by treatment for hypertension slightly attenuated the associations between MCS All and systolic BP (β-coefficient [95%CI]: −2.38, [−4.75, −0.01]) and in MCS with no comorbid FSD the association for systolic BP was no longer significant (β-coefficient [95%CI]: −2.80, [−5.80, 0.19]).

For physical performance, we found a significant lower handgrip strength in MCS All compared with controls, which was not seen in MCS individuals without comorbid FSD. However, both MCS individuals with and without FSD comorbidities had significantly lower step METs count compared with controls which was not fully explained by lifestyle or mental health factors (Table 3). Up to ¼ of participants were unable to complete the step test, that is *n* = 51 (27%), *n* = 20 (18%), and *n* = 1010 (13%) were missing in MCS All, MCS with no comorbid FSD, and the control group, respectively. Among those with missing step test, the majority were women; 73% in MCS All which was significantly more (*p* < 0.0001) than in MCS with no comorbid FSD (60%) and the control group (51%). Finally, odds of high self-perceived fitness were significantly lower among MCS All compared with controls, but not in MCS with no comorbid FSD (Figure 1).

There was no significant effect modification of Cohens perceived stress in any of the outcomes (Appendix A).

## 4. Discussion

To our knowledge, no other studies have previously characterized cardiorespiratory health or physical performance based on clinical objective measures in MCS individuals compared with the general population. In this study, we report a less optimal body composition, impaired cardiorespiratory fitness, and physical performance among MCS individuals compared with the general population. This could to a large extent be explained by coexistence of comorbid FSDs except for step METs count. None of the mental health measures or other included lifestyle factors could explain the observed associations.

Sedentary lifestyle is one of several recognized risk factors for obesity [32]. In a community-based study estimating an individual physical activity coefficient, 27.1% of 52 MCS individuals were categorized as sedentary and 60.4% were categorized as inactive [12]. A previous DanFunD study found lower self-reported physical activity level among MCS individuals suffering from FSD comorbidities compared with the general population [6]. This corresponds to our findings both of a lower self-experienced physical fitness and of a less optimal body composition in terms of higher fat percentage and BMI. Moreover, we found that waist circumference was up to 4 cm higher in MCS compared with controls also illustrated by a higher proportion of MCS exceeding the American Heart Association cutoffs of optimal waist circumference of 102 cm for men (42% vs 25%) and 88 cm for women (40% vs. 29%) [40]. Finally, we found higher risk of MCS individuals with comorbid FSDs being overweight compared with controls. Thus, MCS individuals with FSD comorbidities may be more physically limited resulting in less optimal body composition. A smaller clinical Spanish study (*n* = 52 MCS patients, mean age 50.9 years, 93.2% women), found one third had low muscle strength and 84% has decreased muscle mass (below 10 percentile) [18]. Other studies did not find differences in physical measures such as hand grip and objective measures of physical activity among MCS individuals with chronic fatigue syndrome and fibromyalgia (*n* = 19) (recruited by physicians, media, friends and family), compared with individuals suffering only from chronic fatigue syndrome (*n* = 50) [41]. However, direct comparison cannot be made due to differences in comparison groups. The fact that one fourth of MCS individuals with FSD comorbidities in our study were unable to complete the step test, supports some level of physical disability in of MCS individuals. Missing data may also have attenuated observed associations because the most limited individuals are most likely not represented in the analyses. On the contrary, though MCS individuals without FSD comorbidities had significantly lower step Mets count compared with controls, body composition measures were not significant different from controls.

Lung function (FVC and FEV1) was significantly lower in MCS with FSD comorbidities compared with controls which could not be explained by lifestyle, mental health factors or by OPD and asthma. Because of a relatively big overlap in self-reported information on smoking habits and OPD diagnosis, OPD can to some extend be regarded as reliable but similarly to the asthma diagnosis, this has not been confirmed by reviewing medical records. Previous population-based studies reported associations between MCS and asthma [5,9,19,20,21], however, also based on self-reported MCS. Nevertheless, MCS participants may have symptoms and experience limitations similarly to asthma patients [42] which has been suggested to be related to a lower threshold for coughing and other symptoms that can be induced by chemical activation of the vagus nerve [43]. It should also be noted, that a higher proportion of individuals in MCS All were categorized as overweight/obese (59% vs 43% among controls), which has been shown to be associated with decreased lung function [44].

The results indicate that MCS individuals with no FSD comorbidities can maintain a lifestyle more like the general population than MCS individuals who suffer from one or several other FSD comorbidities. With avoidance of triggering agents being one “treatment” of MCS, it is likely that individuals with MCS and FSD comorbidities in general decrease their engagement in physical activities such as team sports, fitness classes, jogging, daily longer walks etc. to avoid contaminating substances and pain [12]. In line with this, 52 MCS patients recruited from an association of patients with chronic fatigue syndrome and MCS, reported lack of energy as the main cause of limited physical activity (86.5% of 52 patients) [12]. Additionally, as association between leisure time physical activity and hand grip strength has previously been reported in the general population [45] and may also be the case in MCS individuals.

Whether the observed associations are a consequence of MCS is beyond the scope of this paper to confirm and cannot be elucidated based on cross-sectional data. Notably, new theories including environmental chemicals, stress, immunological alterations etc. which overlap with triggering agents and symptoms of MCS have emerged to explain the obesity epidemic [46]. Nevertheless, physical activity could be incorporated in treatment and recommendations for MCS individuals to potentially increase physiological wellbeing and would include long-term health benefits [47].

### Strengths & Limitations

Some strengths and limitation may be considered. The DanFunD cohort constitutes nearly 10.000 adults based on a random sample from the general Danish population [22]. All participants had a general health examination (i.e., physiological health etc.) conducted based on highly standardized protocols by trained nurses and completed comprehensive questionnaires covering a wide range of health information including social factors, occurrence of chronic diseases, mental health etc. Thus, the DanFunD database holds extensive information and is the largest of its kind to study FSDs [22]. However, a participation rate of 34% in DanFunD could question the representativeness of the cohort and participants in this study may be healthier than the non-participants, at least at group level. This was exemplified in a recent study of the DanFunD cohort, where the prevalence of FSD and common mental disorders were found to be higher among non-responders [11]. Still, this would merely underestimate observed associations if non-participants have higher symptom load. Both MCS and other FSD case status were based on questionnaire data and since MCS is rather poorly captured in registries the validated and previously applied case definition is still one of the most reliable tools. It could be suggested, that case status could be improved if clinically verified, however, a DanFunD study concluded that FSD cases identified via self-reported questionnaires compared with FSD cases clinically verified can serve as a suitable screening tools [48]. It is also a limitation that all psychometric parameters are based on self-completed questionnaires and not a clinically diagnosis and that all lifestyles’ parameters such as overall health, comorbidities, sleep and alcohol consumption are self-reported and potentially affected by memory bias

To confirm the observed association between MCS and lung function, clinical verification of lung function, OPD, and asthma should be considered. The step test was most often omitted due to low physical capacity and by more women than men. Since women is often overrepresented in MCS, the results of the step test may not reflect the physical performance among MCS women in general. The number of missing data potentially decreased the real difference between groups and observed associations may not reflect the true level of physical performance in MCS.

## 5. Conclusions

We found less optimal body composition, decreased cardiorespiratory fitness and physical performance among MCS individuals with comorbid FSDs compared with the general population. This was not seen to the same extend in MCS individuals without FSD’s. The degree of symptoms in MCS may impact the possibility to maintain an optimal lifestyle with respect to physical activity which potentially have negative impact on body composition and lung function. None of the mental health measures or other included lifestyle factors could explain the observed associations.

## Figures and Tables

**Figure 1 ijerph-19-09039-f001:**
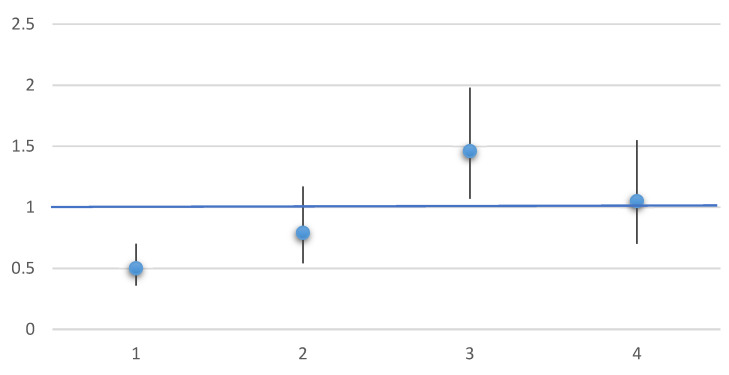
Odds ratio of high self-perceived fitness and overweight in MCS All or MCS with no comorbid FSD. 1: OR (95%CI) of **high self-perceived fitness** in MCS All adjusted for sex and age. 2: OR (95%CI) of **high self-perceived fitness** MCS with no comorbid FSD adjusted for sex and age. 3: OR (95%CI) of **BMI > 25 kg/m^2^** in MCS All Adjusted for sex, age, and age^2. 4: OR (95%CI) of **BMI > 25 kg/m^2^** in MCS with no comorbid FSD Adjusted for sex, age, and age^2. FSD Functional Somatic Disorder. High self-perceived fitness: participants who answered “good” or “very good” on a 5-point scale for self-perceived fitness.

**Table 1 ijerph-19-09039-t001:** Participant characteristics and distributions (mean (SD) is presented unless otherwise stated).

	MCS All (*n* = 188)	*p* ^a^	MCS with No Comorbid FSD (*n* = 109)	*p* ^a^	Controls (*n* = 7791)
Age	53.6 (13.5)	0.21	54.8 (13.6)	0.04	52.7 (13.1)
Sex (% women)	67	<0.001	61	0.03	51
**Anthropometry**					
Waist circumference (cm)	91.6 (16.0)	0.007	90.3 (14.3)	0.32	88.9 (13.3)
Men >102 cm (%) ^b^	42	0.002	36	0.10	25
Women >88 cm (%) ^b^	40	0.009	33	0.48	29
Fat percentage	33.2 (9.0)	<0.001	31.7 (8.7)	0.003	29.1 (8.9)
BMI (kg/m^2^)	27.3 (5.7)	<0.001	26.3 (4.9)	0.35	25.9 (4.4)
% Normal weight	40.9	<0.001	46.8	0.14	47.0
% Overweight	30.8		29.3		37.2
% Obese (class I–III)	28.2		23.8		15.7
**Cardiorespiratory fitness**					
Forced Vital Capacity (L), median (IQR)	3.55 (1.08)	<0.001 ^Ŧ^	3.60 (1.19)	<0.001 ^Ŧ^	3.98 (1.43)
Forced Expiratory Volume s1 (L), median (IQR)	2.73 (0.79)	<0.001	2.74 (0.74)	<0.001 ^Ŧ^	3.07 (1.14)
FVC (L)/FEV1 (L)	0.76 (0.09)	0.06	0.76 (0.08)	0.30	0.76 (0.10)
Systolic blood pressure (mmHg)	127.2 (17.9)	0.05	127.5 (17.5)	0.17	129.8 (18.2)
Diastolic blood pressure (mmHg)	78.4 (10.0)	0.55	78.7 (10.2)	0.90	78.8 (10.3)
**Physical performance**					
Hand grip test (kg), median (IQR)	63.0 (31.0)	<0.001 ^Ŧ^	67.2 (31.7)	<0.001 ^Ŧ^	79.1 (24.7)
Step test (METS count)	8.2 (2.3)	<0.001	8.5 (2.3)	<0.001	9.7 (2.6)
Self-perceived fitness (%good/very good)	27	<0.001	37	0.20	44
**Mental health factors**					
Cohens Stress scale, median (IQR)	13.0 (9.0)	<0.001 ^Ŧ^	12.0 (8.5)	0.001 ^Ŧ^	9.0 (8.0)
Self-perceived health (% good/very good)	27	<0.001	38	<0.001	56
SCL Anxiety, median (IQR)	4.0 (6.0)	<0.001 ^Ŧ^	2.0 (4.0)	<0.001 ^Ŧ^	1.0 (2.9)
SCL Depression, median (IQR)	6.0 (10.5)	<0.001 ^Ŧ^	4.0 (7.0)	<0.001 ^Ŧ^	2.0 (4.9)
**Other lifestyle factors**					
Wake up early (% yes, often)	69	<0.001	43	0.01	33
Cannot sleep (% yes, often)	57	<0.001	18	0.21	14
Alcohol intake					
≤recommended for women, % ^c^	89	0.64	83	0.008	92
≤recommended for men, % ^c^	85	0.17	82	0.42	87
Smoking (yes, daily, or frequently)	17	0.65	17	0.55	15
Self-reported OPD diagnosed by MD, %	9	<0.001	8	<0.001	2
Self-reported asthma diagnosed by MD, %	26	<0.001	18	0.001	9

^a^ Tested mean/median differences between MCS case status and controls using Kruskal-Wallis test for skewed variables ^Ŧ^ otherwise *t*-test. Chi Squared was applied for categorical variables. ^b^ Adapted based on recommended cutoff from Ref. [40]. 2021. American Heart Association. ^c^ Adapted based on recommended cutoff from Ref. [37]. 2010. The Danish National Board of Health. FSD Functional Somatic Disorder, FVC Forced Vital Capacity, FEV1 Forced Expiratory Volume first second, IQR Inter quartile range, OPD Obstructive pulmonary disease MD Medical doctor.

**Table 2 ijerph-19-09039-t002:** Association between anthropometry measures and MCS case status compared to controls.

	β Coefficient (95% CI)		
	Model 1	Model 2	Model 3
**Controls**	1.00	1.00	1.00
**Waist circumference (cm)**			
MCS All	**4.28 (2.65, 5.90)**	**4.23 (2.52, 5.94)**	**3.81 (2.09, 5.54)**
MCS with no comorbid FSD	2.05 (−0.06, 4.16)	2.26 (0.07, 4.45)	2.07 (−0.11, 4.26)
**Body fat (%)**			
MCS All	**2.13 (1.18, 3.09)**	**2.25 (1.24, 3.26)**	**2.02 (1.00, 3.04)**
MCS with no comorbid FSD	1.02 (−0.21, 2.27)	1.18 (−0.10, 2.48)	1.08 (−0.20, 2.38)
**BMI (kg/m^2^)**			
MCS All	**1.53 (0.90, 2.16)**	**1.50 (0.85, 2.17)**	**1.36 (0.70, 2.03)**
MCS with no comorbid FSD	0.49 (−0.33, 1.30)	0.56 (−0.28, 1.40)	0.49 (−0.35, 1.34)

FSD Functional Somatic Disorder. Model 1: adjusted for sex, age, and age^2. Model 2: additionally, adjusted for alcohol, smoking, and sleep disturbances. Model 3: additionally, adjusted for Cohens perceived stress scale, depression, and anxiety.

**Table 3 ijerph-19-09039-t003:** Association between cardiorespiratory fitness, physical performance and MCS All or MCS with no comorbid FSD compared to controls.

	β Coefficient (95% CI)		
	Model 1	Model 2	Model 3
Cardiorespiratory fitness			
**FVC *(L)**			
MCS All	**−0.21 (−0.29, −0.13)**	**−0.20 (−0.29, −0.12)**	**−0.20 (−0.27, −0.11)**
MCS with no comorbid FSD	**−0.12 (−0.23, −0.02)**	**−0.09 (−0.20, 0.02)**	**−0.08 (−0.19, 0.02)**
**FEV1 *(L)**			
MCS All	**−0.29 (−0.27, −0.12)**	**−0.18 (−0.26, −0.09)**	**−0.17 (−0.25, −0.09)**
MCS with no comorbid FSD	**−0.11 (−0.21, −0.02)**	**−0.08 (−0.18, 0.02)**	**−0.07 (−0.17, 0.03)**
**FEV1/FVC**			
MCS All	−0.01 (−0.02, 0.00)	−0.01 (−0.03, 0.00)	−0.01 (−0.03, 0.00)
MCS with no comorbid FSD	−0.008 (−0.02, 0.01)	−0.006 (−0.02, 0.01)	−0.006 (−0.03, 0.01)
**Systolic BP (mmhg)**			
MCS All	**−2.54 (−4.83, −0.25)**	**−2.40 (−4.83, 0.03)**	**−2.66 (−5.11, −0.21)**
MCS with no comorbid FSD	**−3.48 (−6.46, −0.50)**	**−3.86 (−6.97, −0.75)**	**−3.93 (−7.03, −0.82)**
**Diastolic BP (mmhg)**			
MCS All	0.12 (−1.25, 1.51)	0.34 (−1.12, 1.82)	0.19 (−1.29, 1.68)
MCS with no comorbid FSD	0.05 (−1.74, 1.85)	−0.02 (−1.91, 1.85)	−0.08 (−1.97, 1.80)
Physical performance			
**Handgrip (pounds)**			
MCS All	**−5.64 (−7.70, −3.58)**	**−4.54 (−6.71, −2.36)**	**−3.98 (−6.18, −1.78)**
MCS with no comorbid FSD	−2.42 (−5.10, 0.25)	−1.88 (−4.68, 0.90)	−1.63 (−4.42, 1.16)
**Steptest (Mets)**			
MCS All	**−1.20 (−1.59, −0.82)**	**−1.17 (−1.57, −0.77)**	**−1.07 (−1.47, −0.67)**
MCS with no comorbid FSD	**−0.70 (−1.18, −0.22)**	**−0.73 (−1.22, −0.24)**	**−0.68 (−1.16, −0.19)**

FSD Functional Somatic Disorder, FVC Forced Vital Capacity, FEV1 Forced Expiratory Volume first second, BP Blood pressure. Model 1: adjusted for sex, age, and age^2. Model 2: additionally, adjusted for alcohol, smoking, and sleep disturbance. Model 3: additionally, adjusted for Cohens perceived stress scale, depression, and anxiety. * FVC and FEV1 also adjusted for height in all models.

## Data Availability

Data cannot be made publicly available for ethical and legal reasons. Public availability may compromise participant privacy, and this would not comply with Danish legislation. Access to the subset of data included in this study can be gained through submitting a request to The Capital Region Knowledge Center for Data Compliance, The Capital Region Denmark; cru-fp-vfd@regionh.dk. Acquisition of data are only allowed after permission to handle data has been obtained in accordance with the guidelines stated by the Danish Data Protection Agency: http://www.datatilsynet.dk/english (accessed on 1 February 2022).

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
