# Peer review of "Physiological Health and Physical Performance in Multiple Chemical Sensitivity—Described in the General Population"

_ijerph, 2022, doi:10.3390/ijerph19159039_

Round 1

Reviewer 1 Report

Authors have addressed most of my comments. I have no further comments.

Kind regards,

This manuscript is a resubmission of an earlier submission. The following is a list of the peer review reports and author responses from that submission.

Round 1

Reviewer 1 Report

This is a very interesting study, but it requires clarifying some methodological aspects to validate its results.

The N= 9,556, of these one (n)= 188 was obtained, of which 109 without other somatic disorders. The rest was considered a control group (?) How much are the controls finally. In these cases, it is advisable to make a flowchart to better illustrate the selection process and the characteristics of the research subjects.

It should be better described what the final controls are. When reviewing the sample, it is understood that 188 have MCS and of these 188, 109 were categorized as FSD

Therefore, in this case what has some MCS would be 297 MCS +FSD. But it is stated that the rest became controls or 9256 controls (?).

This is important to clarify because it is understood that this group of subjects is a sample of a sample and therefore care must be taken of the effect on the statistical power and the inferences made and if these will be possible to extrapolate to the population.

It is stated that there are no clear criteria defined with a wide prevalence range since there is no adequate criterion to define MCS, it is stated that the reference establishes prevalence’s between 0.5% and 12% and that many times it is not even diagnosed.

Regarding the above, the aim of the article was to further elucidate whether MCS is associated with poor physiological health in terms of poor physical performance, low cardiorespiratory fitness, and optimal body composition. While this knowledge may contribute to focused health promotion among individuals and patients with MCS.

However, the question arises if it is possible to associate. In the first place, it is important to address theoretical aspects of the associations made on MCS (outcome) and the covariates. Second, because it is an instrument that collects past information. Therefore, there is a risk of memory or a risk of self-selection of those who answer the survey since they can identify with the health problem for their own interest or that of a family member. What was the mechanism used to control this bias and other biases that might have arisen? The three criteria used for classification do not explain how they control the possible biases mentioned above. Unless patients with MCS are diagnosed a priori.

Reviewer 2 Report

The work is well written and objective. The methods are appropriate and well described. I suggest minor adjustments as described below.

• Adjust terms used to refer to groups (for example, MCS ÷other FSD). It's confused!

• In the results, there is much repetition in the text of the table values. That's not necessary. Just do an analysis.

•With more than 9 thousand participants, the on-site assessments must have taken time. I suggest recording this time interval.

• Some references are very old. I suggest using them only in case they are exclusively necessary. If not, update them.

Reviewer 3 Report

Dear author, first of all thank you for your submission. The article aims to explore the associations between Multiple chemical sensitivity and objective measures of anthropometry, cardiorespiratory health and physical performance presented is of superior quality and I believe it can be a manuscript cited for many years. The authors must be congratulated and I only ask that they can review the fields that are too many, as in the first line of the abstract. In addition, they could perhaps choose to display the results by p-values with only 3 decimal places following the guidelines for statistical reporting (APA) or justify the reason why they are presented in the current form.

Reviewer 4 Report

Thank you so much for inviting me to review this manuscript. Authors tried to "elucidate whether MCS is associated with poor physiological health in terms of poor physical performance, low cardiorespiratory fitness, and less optimal body composition".  The article deals with an interesting subject and includes a wide sample. However, I have some comments and concerns that prevent me from accepting the manuscript. 

Major comments:

- Why did the authors choose these covariates and not others? This should be justified on the basis of the literature. 

- A flow chart is necessary for a better understanding of the selection process of the sample.

- Were all assumptions for linear regression met (normality, heteroscedasticity, etc.)? If there are variables with non-normal distributions, what have the authors done to introduce them into the models? Please, provide information about this issue.

- Authors must include the p values of the interactions (sex and age). Although at the statistical level there was no interaction in relation to sex or age. At the clinical level, there are large differences in the age range 18 to 76 years, as well as between men and women. I believe this is the major limitation of this manuscript. 

- There are other relevant limitations that the authors have not indicated, such as the design of the study or some of the variables obtained. 

Minor comments:

- Abstract should include the aim of the study.

- The references are not in the journal format.

- In the authors' contributions, indicate only your initials.

- Waist circumference was measured only once. What about technical measurement error? 

- "Poor" physical performance is a value judgment. What is "poor"? A different term (e.g., lower) would be more appropriate. 

Best wishes,

Reviewer 5 Report

I have read this paper with interest. The shape of the article is correct, well writen and presents lots of data in an ordered manner. The authors have developed a good work and this article will suitable for publication in this journal after minor changes are done.

The introduction of this paper is well writen and introduces the concept of MCS properly. However, i am missing a paragraph summarizing the main effects of exercise on people with MCS both acute and chronic.

In line 55 you use the abbreviation DanFunD but you don't define it until line 85. Please, mention in L55.

L185. Please, define Standard deviation (SD) before using the abbreviation.

L161 indicate One-way ANOVA if it was the test you used.

L185-187 The information in this line is duplicated in the table. Delete from one of the sources.

I don't understand the use of p values in table 1 for the variables expressed as % such us sex, Men>102 cm... Perhaps you could use a symbol to remark the use of chi squared
